# Digital Technology and Innovative Technology to Promote the Professional Development of Digital Media Based on Green Energy under COVID-19

Qianqian Xu [1,2], Bing Zheng [3,*], Hongmi Zhou [2], Jingfan Chen [4], Zhifeng Zhang [5] and Xueping Wu [1]

1    College of Education, Zhejiang University, Hangzhou 310027, China
2    Hangzhou Electronic Information Vocational School, Hangzhou 310021, China
3    School of Business Administration, Zhejiang Gongshang University, Hangzhou 310018, China
4    School of Art and Design, Zhejiang Business College, Hangzhou 310053, China
5    Department of Mechatronics Engineering, Zhejiang University, Hangzhou 310027, China
*    Correspondence: zerobeat@zjgsu.edu.cn

**Abstract:** Taking as an example the practical teaching of the design of children's solar-energy-based ultraviolet disinfection products, we analyzed the practical activities in four stages of teaching—case background, research methods, product design, and practical results—in the practical teaching mode based on solar green energy. This paper presents and proposes a design solution for a solar-powered green energy-based multifunctional inductive UV disinfection product for children to provide additional services for school interventions and improve public health in primary and secondary schools. This new innovative design for a children's disinfection product is based on solar green energy and enhances the graded disinfection strategy in schools, reducing the number of viruses and the potential risk of virus transmission in the educational environment. The proposed program aims to be project-oriented, combining green energy concepts with innovative educational concepts, classroom content with social prevention products, digital technology with innovative thinking, promoting the development of innovative and digital abilities of teachers and students, and promoting the development of practical teaching in digital media. The practical results show that the model has positive teaching effects, practical value for students, schools and society, cultivation of digital innovation ability of teachers and students, and reference significance for practical teaching.

**Keywords:** digital media major; practical innovation; solar energy; green energy; disinfection products; UV disinfection; AHP; PUGH





## 1. Introduction

The teaching of digital media is an emerging profession that combines digital technology, art, and innovation. It requires not only a solid theoretical foundation but also a high level of practical creative ability. As a subject requiring a strong knowledge of digital technology, strong practicality, and strong innovation ability, practical teaching plays a key role in improving the quality of digital media teaching and enhancing students' abilities. At present, on the one hand, the practical teaching of digital media is biased towards theoretical teaching and simple technical operation, ignoring the application of innovative technology and digital technology in practical teaching. On the other hand, there is a certain degree of disconnection between the practical teaching of digital media and the actual production in the market and industry, and it lacks relative independence, systematization, continuity, and mutual coordination. In the social environment of COVID-19, a project-oriented approach is needed to combine green energy concepts with innovative educational concepts, classroom content with social prevention products, and digital technology with innovative thinking to promote the development of innovative and digital skills in teachers and students, as well as to promote the development of practical digital media teaching.

This paper introduces and proposes a design scheme of a multifunctional children's induction UV disinfection product based on solar green energy. Taking as an example the practical teaching of the design of children's solar-energy-based ultraviolet disinfection products, we analyzed the practical activities in four stages of teaching— case background, research methods, product design, and practical results—in the practical teaching mode based on solar green energy.

At present, children are increasingly vulnerable due to the COVID-19 pandemic. As places with high populations of children, schools are prone to high levels of viral cross-infection. Schools play an important role in the lives of children and their families. Therefore, every reasonable effort should be made to keep the school safe and open. To achieve safer face-to-face learning, schools should implement disinfection strategies as far as possible to protect students from COVID-19, thereby reducing the spread of COVID-19 in schools. Since infected bacteria can live in a person's body and surroundings, more effective interventions should encourage both personal hygiene (such as hand hygiene) and environmental disinfection (such as cleaning surfaces).

Frequent disinfection is one of the best ways to effectively control the spread of an outbreak. During the outbreak of a virus such as COVID-19, in order to reduce crossbreeding, the common disinfection methods in a school's public places are spraying disinfectant, providing hand disinfectant, and ultraviolet or physical cleaning [1]. Research in the literature has found that school disinfection methods currently focus on hand hygiene. Researchers in San Francisco found that promoting mitigation policies can effectively prevent the spread of COVID-19 in educational settings. The promotion of mitigation policies mainly includes wearing masks, maintaining hand hygiene in rooms, etc. [2] Studies have referred to good hand hygiene as necessary to control and prevent infection, but many children do not wash their hands adequately. While there is classroom communication for children and available toilet space, many places for hand hygiene activities are ignored [3]. A prototype of a smart handwashing station deployed in a school setting during the COVID-19 pandemic was tested. A personalized approach was used with technology proven to be successfully implemented in schools to improve children's hand hygiene [4]. Some studies have adjusted the COVID Tracer advanced tool of the Centers for Disease Control and Prevention of the United States to model the mitigation strategy of returning to school during the COVID-19 pandemic. The study shows that implementing and strictly adhering to key mitigation strategies such as cleaning and disinfection, hand hygiene, etc., can reduce transmission rates in schools. The research team conducted a study on the disinfection of 32 primary and secondary school situations in the Zhejiang region of China. According to the survey records, 100% of the schools used a variety of disinfection supplies, such as ether, 75% ethanol, chlorinated disinfectants, disinfection solutions such as peroxyacetic acid and chloroform, and ultraviolet devices. It was observed that 75% ethanol disinfectant solutions were placed at the class entrances or washroom sinks in 100% of primary and secondary schools, but these disinfectant solutions had contact press pumps, which were prone to cross-infection. It was also found that children's pencils, erasers, and other school utensils did not have targeted disinfection devices. Washing clean hands and touching germ-ridden press pumps or school utensils could easily re-infect them with bacteria and viruses.

While chlorinated sanitizers or PCR cleaner solutions are also quick and effective germicidal solutions, they tend to have the negative features of pungent odors, corrosiveness, bleaching properties, and chemical insensitivity. In addition, they cause potential toxic damage to ecosystems and their use is detrimental to the environment. Several studies have shown that large doses and high concentrations of chlorinated disinfectants can cause potentially toxic damage to ecosystems. However, ultraviolet light has the advantages of high-efficiency sterilization, no secondary pollution, complete environmental protection, and no drug resistance. Ultraviolet rays directly damage DNA, RNA, proteins, etc., in bacterial virus cells, causing direct cell death and the inability to reproduce and replicate, and there is no drug resistance. The whole sterilization process does not require additional chemicals, no other chemical pollutants are produced, and there is no residual secondary

pollution, which truly achieves complete environmental protection. Specifically, previous studies had found that UV radiation emitted at 254 nm is the most effective [5]. Most UV indoor disinfection equipment uses mercury gas bulbs as the light source with a characteristic wavelength of 254 nm [6].

A team of researchers applied different disinfection methods to observe the effectiveness of the removal of COVID-19 nucleic acid contamination from plastic surfaces to test the effectiveness of different methods of disinfection. The following disinfection effects were found: 2000 mg/L chlorinated disinfectant = 5500 mg/L chlorinated disinfectant > 750 m L/L ethanol > PCR cleaner. In addition, the team tested the effectiveness of different UV exposure times on the removal of COVID-19 nucleic acid contamination and the following disinfection effects were found: 3 h of UV exposure = 4 h of UV exposure = 5 h of UV exposure > 1 h of UV exposure > 2 h of UV exposure. The above experimental data led to the conclusion that 2000 and 5500 mg/L chlorinated disinfectants and 3 h of UV exposure were the most effective in removing nucleic acid contamination [7].

At the same time, raw material supplies are drying up as population and energy demands grow. In response to these challenges, the transition to sustainable energy is taking place around the world today. Against the background of the worldwide "Carbon Double", the development of renewable energy [8], like solar energy, has received unprecedented attention. Due to its continuous fusion reaction, the sun is a super-abundant source of permanent energy. On Earth, solar panels consume a limited amount of energy through photovoltaic technology. This technology plays an important role in combating global warming and meeting future energy needs. In addition to zero emissions, Solar energy also contributes to reducing the carbon footprint by directly reducing greenhouse gas emissions [9,10]. Therefore, against the background of COVID-19, there is an urgent need for a UV disinfection product for children based on solar green energy.

This paper presents a solution for the design of a multifunctional inductive children's UV disinfection product based on solar green energy. The aim of the proposed solution is firstly to provide a new and innovative design for children's disinfection products, bridging the research gap in the disinfection of students' tools in order to improve public health in primary and secondary schools and to reduce the potential risk of virus transmission in the educational environment. The design of the disinfection product is based on solar green energy, taking advantage of the sustainable development of green energy to address the current energy shortage and keep the natural ecosystem in a virtuous cycle. Moreover, this integrated practical model helps students to fully integrate technical means, artistic creation, and creative practice, giving full play to student motivation in the creative activities of social innovation practice and making teaching and learning more practically meaningful.

## 2. Research Methodology

### 2.1. Research Methodology Principles

#### 2.1.1. Principle of AHP

In the early stages of product development, when multiple criteria exist, the situation becomes complex and the research team's ability to select the influencing factors for the product becomes important. The team needs to compare the various influencing factors for the product, derive a weighted ranking based on the importance of the factors and then carry out a quantitative analysis to compare them. This requires formal methods to ensure a structured means of decision making. AHP is a widely used multicriteria decision-making method based on a structured two-by-two comparison of alternative criteria and prioritization weights [11]. The AHP constructs problems in a hierarchical manner, descending from an objective into successive levels of criteria, subcriteria, and alternative factors [12]. By constructing an AHP hierarchical evaluation model and an AHP judgement matrix, the influencing factors of the product are hierarchical and data-driven, and the weights of the influencing factors are obtained, making it easy to make more accurate decisions in the later PUGH matrix. AHP provides the overall view for

the research team, and the weights of the influencing factors of the design products are obtained by pairwise comparison of the nine grade scales.

The broad steps in the process of AHP application include the following three aspects. ① Constructing the AHP hierarchical evaluation model which is mainly used to confirm the target layer, criterion layer, and judgment layer of the whole design event. ② Constructing the judgement matrix by comparing the factors with each other two by two and determining the weight of each judgement layer to the target layer. ③ Consistency testing which is used to determine whether there are logical problems with the constructed judgment matrix.

2.1.2. Principle of PUGH

In the product development process, the ability to make decisions about design principles is particularly important when teams are faced with multiple design options. The research team needs to compare and weigh up the design options in order to select the optimal design solution. The proper use of trade-off analysis tools can provide a formal and scientific approach to the solution selection process. The most common method currently used is the PUGH matrix selection method. The idea screening and selection methods proposed by Dr. Stuart Pugh are widely used. The PUGH matrix is a well-established tool that is particularly useful for evaluating different ideas in the early stages of product design as it assesses the overall concept from several aspects. It is particularly useful when a suitable product already exists as this will serve as a benchmark for other concepts to be evaluated [13]. It is also simple and straightforward to use. By establishing evaluation criteria and constructing a decision matrix, the alternatives or designs are compared with the current data to help the research team select the optimal solution [14]. This decision is based on whether the optional alternatives or designs are equal, inferior, or superior to the benchmark [15].

PUGH can assist designers in the following ways. ① It can perform a conceptual screening of various possible conceptual design solutions, using concept scoring and ranking them to select the best solution that meets the requirements of the demand specifications. ② It can enhance the concept design by integrating various criteria.

The PUGH matrix analysis has a significant impact on the success of early projects, enabling quick and correct decisions to be made in the face of the many options available. In product concept design, the design team first evaluates the many options internally and identifies three to four creative directions according to the needs of the project. The team leader, structural engineer, and industrial designer then form an evaluation team to score, rank, select, and weigh up the 3–4 options. The PUGH matrix analysis provides the basis for the development of the product. It enables the qualitative comparison and judgement of multiple design solutions and can help teams to quickly filter out the more obvious winning solutions or promising solutions for further observation and comparison. It provides the basis for product development, and it can result in at least four kinds of decisions [16]:

- Remove certain weak concepts from consideration.
- Contribute to the further development of certain concepts.
- Contribute to design information gathering.
- Develop other concepts based on what is revealed through the matrix and the discussions it provokes.

*2.2. Research Steps*

2.2.1. Model Construction

Based on the product design methodology and the user requirements questionnaire, and after expert analysis, the research team constructed a hierarchical demand analysis model for children's disinfection products, as shown in Figure 1.

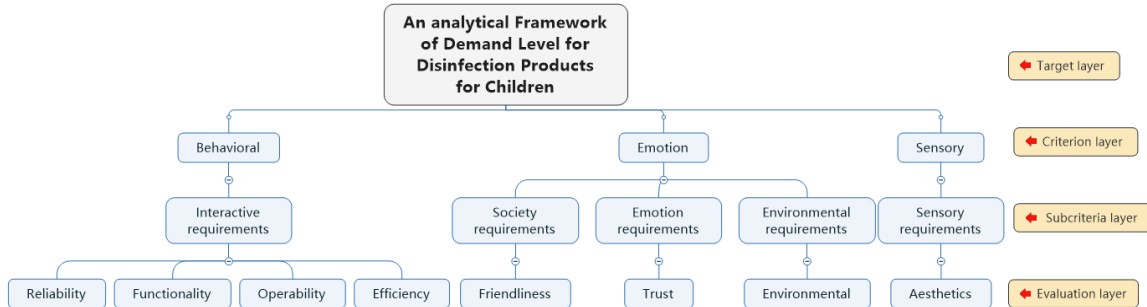

**Figure 1.** Hierarchical analysis model for children's disinfection products.

Behavioral experience, which focuses on the interactive needs of children's disinfection products, includes the reliability of the disinfection effect of the product, the perfection of the product's functional design, operability, and the efficiency of killing toxins.

Emotional experience, which is an extension of behavioral experience, focuses on users' social, emotional, and environmental needs, mainly referring to users' needs for children's disinfection products.

Sensory experience—a user's sensory needs, including sight, smell, hearing, and touch—is for the most part mainly a user's audio–visual experience, with an emphasis on comfort. Children's disinfection products are mainly concerned with the aesthetics of the product shape, that is, the aesthetic experience of the visual sense, and the product design style should be in line with the aesthetic orientation of children.

### 2.2.2. Judgement Matrix and Weights

The research team collected the scores of experts and users, made a two-by-two judgment on the indicators in the judgment layer of the hierarchical analysis model, and used the Santi 1–9 scale method to give a judgment scale. The higher the value obtained for this indicator, the higher its relative importance. The team then ran the indicators by constructing an AHP judgment matrix for children's disinfection products, as shown in Table 1.

**Table 1.** AHP judgement matrix.

| Target Layer A | Evaluation Layer $B_1$ | Evaluation Layer $B_2$ | Evaluation Layer $B_3$ | ... | Evaluation Layer $B_N$ |
|---|---|---|---|---|---|
| evaluation layer $B_1$ | 1 | $a_{12}$ | $a_{13}$ | | $a_{1N}$ |
| evaluation layer $B_2$ | $a_{21}$ | 1 | $a_{23}$ | | $a_{2N}$ |
| evaluation layer $B_3$ | $a_{31}$ | | 1 | | $a_{3N}$ |
| . . . | . . . | . . . | . . . | . . . | . . . |
| evaluation layer $B_N$ | $A_{N1}$ | $A_{N2}$ | $A_{N3}$ | | 1 |

The calculation formula is as follows:

$$a_{ij} = b_i/b_j \quad b_{ij} = c_i/c_j \tag{1}$$

The research team calculates the mth power of the product of each row to obtain the m-dimensional vector, as shown in Formula (2).

$$\overline{W_j} = \sqrt[m]{\prod_{j=1}^{m} a_{ij}} \tag{2}$$

Then, the vector is normalized to a weight vector, which gives the weights, as shown in Formula (3).

$$W_i = \frac{\overline{W_i}}{\sum_{j=1}^{m} \overline{W_j}} \tag{3}$$

Based on the above weight solution method, the AHP judgement matrix for children's disinfection products is solved, together with the weight vectors for the criterion layer and the different evaluation layers. The weights for the criterion layer are shown in Table 2. The weights for each evaluation layer are shown in Tables 3 and 4.

**Table 2.** Weight value of criterion layer under target layer A.

| Target A | Behavior $B_1$ | Emotion $B_2$ | Sensory $B_3$ | Weight Value |
|---|---|---|---|---|
| behavior $B_1$ | 1 | 3.3333 | 8.3333 | 0.7082 |
| emotion $B_2$ | 0.3 | 1 | 2.1739 | 0.2028 |
| sensory $B_3$ | 0.12 | 0.46 | 1 | 0.089 |

**Table 3.** The weight value of each evaluation index under the Behavior criterion layer $B_1$.

| Behavior $B_1$ | Reliability $C_1$ | Functionality $C_2$ | Operability $C_3$ | Efficiency $C_4$ | Weight Value |
|---|---|---|---|---|---|
| reliability $C_1$ | 1 | 3.3333 | 1.1765 | 0.7576 | 1.3129 |
| functionality $C_2$ | 0.3 | 1 | 0.2353 | 0.1905 | 0.3405 |
| operability $C_3$ | 0.85 | 4.25 | 1 | 0.6667 | 1.2457 |
| efficiency $C_4$ | 1.32 | 5.25 | 1.5 | 1 | 1.7956 |

**Table 4.** The weight value of each evaluation index under the Emotion criterion layer $B_2$.

| Emotion $B_2$ | Friendliness $C_5$ | Trust $C_6$ | Environmental $C_7$ | Weight Value |
|---|---|---|---|---|
| friendliness $C_5$ | 1 | 0.5 | 0.7692 | 0.2348 |
| trust $C_6$ | 2 | 1 | 1.25 | 0.4381 |
| environmental $C_7$ | 1.3 | 0.8 | 1 | 0.3271 |

### 2.2.3. Consistency Check and Comprehensive Weights

Let the A–B judgment matrix be A and let the B–C judgment matrix be B. Calculate the maximum characteristic root, as shown in Formula (4).

$$\lambda_{max} = \frac{1}{n} \sum_{i=1}^{n} \frac{(AW)_i}{W_i} \tag{4}$$

At the same time, the consistency of the judgment matrix is checked by quantification, where *CR* denotes the consistency test and *RI* denotes the randomness indicator, as shown in Formula (5).

$$CI = \frac{\lambda_{max} - n}{n - 1} \tag{5}$$

Look up the corresponding RI value according to the RI table. Perform the CR value as shown in Formula (6). If the value of CR is less than 0.1, it indicates that it passes the consistency check.

$$CR = CI/RI \tag{6}$$

Finally, the combined weights are derived. Assuming that the weight of the criteria layer is set to S and the weight of the target layer is $T_j$, the final comprehensive weight $E_i$ is derived, as shown in Formula (7).

$$E_i = W_i \times T_j \tag{7}$$

The Behavior criterion layer B1 judgment matrix is calculated to obtain a CR value of 0.0056 < 0.1. The Emotion criterion layer B2 judgment matrix is calculated to obtain a CR value of 0.0046 < 0.1. The CR value obtained from the judgement matrix of target layer A is 0.0021 < 0.1. Sensory criterion layer B3 does not require additional operations because there is only one indicator. The CR values obtained from Table 5 are all less than 0.1 and pass consistency checks. This conclusion indicates that the degree of consistency of the constructed product design solutions for children's UV disinfection products is found to be within tolerable limits and that there are no logical problems.

**Table 5.** The results of the consistency check.

| Judgement Matrix | $l_{max}$ | CI | RI | CR | Consistency Check Result |
|---|---|---|---|---|---|
| target layer A | 3.0022 | 0.0011 | 0.525 | 0.0021 | passed |
| behavior criterion layer $B_1$ | 4.0147 | 0.0049 | 0.882 | 0.0056 | passed |
| emotion criterion layer $B_2$ | 3.0048 | 0.0024 | 0.525 | 0.0046 | passed |

Once the consistency all passed, the combined weights were then calculated according to Equation (7), as shown in Table 6.

**Table 6.** The results of the comprehensive weights.

| Target Layer | Criteria Layer | Evaluation Index | Weights of Evaluation Index under the Criterion Layer (W) | Weights of Criterion Layer under Target Layer A (T) | Comprehensive Weight (E) |
|---|---|---|---|---|---|
| target A | behavior $B_1$ | reliability $C_1$<br>functionality $C_2$<br>operability $C_3$<br>efficiency $C_4$ | 1.3129<br>0.3405<br>1.2457<br>1.7956 | 0.7082 | 0.9298<br>0.2411<br>0.8822<br>1.2716 |
| | emotion $B_2$ | friendliness $C_5$<br>trust $C_6$<br>environmental $C_7$ | 0.2348<br>0.4381<br>0.3271 | 0.2028 | 0.0476<br>0.0888<br>0.0663 |
| | sensory $B_3$ | aesthetics $C_8$ | 1 | 0.089 | 0.089 |

### 2.2.4. PUGH Matrix Construction

The rating assessment of the PUGH matrix judging layer indicates the difference in relative performance through a score of five bands. A score of −2 means worse than the baseline solution, −1 means slightly worse, S means the same, +1 means slightly better, and +2 means better. The score is calculated by combining the importance of the design criteria, and the final score is given in Formula (7), which is the weight of the ith criterion based on the importance of the design principle and represents the score of solution j on the ith criterion. Finally, the final solution is determined based on the total score, summarizing the design principles of the disinfection product, and enhancing the practicality and user experience satisfaction of the product.

$$R_j = \sum_{i=1}^{n} W_i R_{ij} (j = 1, 2, 3, \ldots, n) \tag{8}$$

In the process of designing the children's disinfection product, the design team developed three alternative initial design options. Based on the above designed PUGH matrix judging criteria and judging formula for the children's disinfection product design options, the current children's disinfection options used in primary and secondary schools were used as the baseline options and the three newly designed options were compared to them to obtain the following PUGH matrix, as shown in Table 7.

**Table 7.** PUGH matrix and scores for children's disinfection product design solutions.

| Evaluation Index (i) | Comprehensive Weight (E) | Datum Scheme | Alternatives (R) | | | Alternatives with Weight | | |
|---|---|---|---|---|---|---|---|---|
| | | | Scheme X1 | Scheme X2 | Scheme X3 | Scheme X1 | Scheme X2 | Scheme X3 |
| reliability $C_1$ | 0.9298 | S | S | +1 | +2 | 0 | 0.9298 | 1.8596 |
| functionality $C_2$ | 0.2411 | S | −1 | S | +1 | −0.2411 | 0 | 0.2411 |
| operability $C_3$ | 0.8822 | S | +1 | S | +2 | 0.8822 | 0 | 1.7644 |
| efficiency $C_4$ | 1.2716 | S | +1 | +1 | +2 | 1.2716 | 1.2716 | 2.5433 |
| friendliness $C_5$ | 0.0476 | S | +1 | +2 | +2 | 0.0476 | 0.0952 | 0.0952 |
| trust $C_6$ | 0.0888 | S | S | +1 | +1 | 0 | 0.0888 | 0.0888 |
| environmental $C_7$ | 0.0663 | S | S | +1 | +1 | 0 | 0.0663 | 0.0663 |
| aesthetics $C_8$ | 0.089 | S | −1 | S | +1 | −0.089 | 0 | 0.089 |
| total score | | | | | | **1.8713** | **2.4519** | **6.7478** |

It can be seen from Table 8 that the X1 sequence of the existing optional scheme is {S, −1, +1, +1, +1, S, S, −1}, and the lowest total score after weighting is 1.8714; the optional scheme X2 sequence is {+1, S, S, +1, +2, +1, +1, S}, and the weighted total score is 2.4519; the option X3 sequence {+2, +1, +2, +2, +2, +1, +1, +1}, and the highest score is 6.7478. After selection and trade-off, according to the weighted scheme score results, the three schemes were better than the currently used benchmark scheme, and the optional scheme X3 had obvious advantages. Therefore, the development and selection scheme X3 of the multifunctional inductive UV disinfection products for children was selected as the basis for further development and design.

**Table 8.** PUGH matrix and score of children's disinfection product design scheme comparison.

| Evaluation Index (i) | Datum Scheme | User Evaluation for Scheme X3 with Weight | | | | | | | | | |
|---|---|---|---|---|---|---|---|---|---|---|---|
| | | User1 | User2 | User3 | User4 | User5 | User6 | User7 | User8 | User9 | User10 |
| reliability | S | +1 | +2 | +2 | +1 | +1 | +2 | +2 | +2 | +1 | +2 |
| functionality | S | +2 | +2 | +1 | +2 | +1 | +2 | +1 | +2 | +1 | +1 |
| operability | S | +2 | +2 | +1 | +2 | S | +2 | +2 | +2 | +2 | +2 |
| efficiency | S | +2 | +2 | +1 | +2 | +1 | +2 | +1 | +2 | +1 | +1 |
| friendliness | S | +1 | +1 | S | +1 | +1 | +1 | +2 | +1 | +2 | +1 |
| trust | S | +2 | +1 | S | +1 | +1 | +1 | +1 | +1 | +2 | +1 |
| environmental | S | +1 | S | +1 | +1 | S | +1 | S | S | +1 | S |
| aesthetics | S | +1 | +1 | +2 | +2 | +1 | +2 | +1 | +1 | +2 | +1 |

*2.3. Results Analysis*

Based on the product design methodology and the user requirements questionnaire, and after analysis by experts, the research team constructed a layered requirements analysis model for a children's disinfection product. The target layer was to design a multifunctional children's disinfection product. The criterion layer included three factors: Behavioral, Emotion, and Sensory. The final eight criteria were Reliability, Functionality, Operability, Efficiency, Friendliness, Trust, Environmental protection, and Aesthetics. This structure was determined to be reasonable by constructing a judgement matrix and consistency testing according to the AHP research method. By comparing the factors two by two, the weights of each judgment indicator to the target layer were determined. Next, the PUGH matrix was constructed according to the PUGH research method to compare the three possible conceptual design solutions, score and rank them conceptually, and select the best solution that meets the requirements of the demand specification X3.

### 3. Product Design

*3.1. Design Principles*

As mentioned earlier, in the study of innovative designs for children's disinfection products, Option X3, the multifunctional, inductive children's UV disinfection product design, was chosen. The design principles are summarized below:

- In terms of green energy, photovoltaic solar panels are used to provide free green electric energy for sterilization and disinfection devices, and the photovoltaic effect of semiconductor materials is used to directly convert solar radiation energy into electric energy.
- In terms of product reliability, the use of constant current drive will make the product start faster and safer. At the same time, the product design pays attention to improving the service life of the product, making it up to 30,000 h, so it can solve the difficulty of sterilizing and disinfecting at every moment.
- In terms of functional perfection, it has kinetic induction, self-help warning, and automatic power-off functions.
- In terms of product operability, it is easy to install and easy to use, without the need for rectifiers and glow starters. Moreover, the cost is low. The performance is improved, while the wear rate is reduced, and maintenance costs are saved. Therefore, it is conducive to the popularization of the disinfection product market.
- In terms of efficiency, the UV light wave penetration rate is higher, the intensity is greater, and the effective sterilization is $\geq$99%.
- In terms of friendliness, the disinfection products are well-suited to the actual needs of the market, reducing the harm caused to children by infectious diseases and bacterial infections, providing automatic warning to reduce the chances of illness, providing a more hygienic and cleaner environment for children's learning environments, and being user-friendly.
- In terms of trust, the UVC products utilize the UVC band with outstanding germicidal effect and the hand sanitizer uses mild ethanol to ensure effective daily disinfection and removal of nucleic acid contamination.
- In terms of environmental protection, this product kills microorganisms using UV light, destroying the nucleic acid structure of the microorganisms, and making them irreproducible so as to achieve the purpose of sterilization. It has no chemical residues or heavy metal residues, making it more environmentally friendly.
- In terms of aesthetics, the main color is white, and the lid is made of transparent material, with a focus on the aesthetics of the product structure, the aesthetics of the process, and the comfort of the material.

*3.2. Kernel Module*

3.2.1. Solar Green Energy Technology Module

Photovoltaic solar panels are used to provide free green electricity for the sterilization and disinfection device. The switch controls the opening and closing of the ultraviolet lamp, which can be moved to any place with sunlight and is not limited by the power grid and the use site. The power socket can use the grid power supply to supplement the power in case of insufficient sunlight.

3.2.2. UV LED Disinfection Technology Module

The interior of the product box is equipped with a UV-LED lamp board, and UV-LED bulbs are distributed on UV-LED installations These UV-LED light panels form a 360-degree UV light surround irradiation system that uses UV light to kill bacteria and viruses on pencils, erasers, keys, and other utensils placed in the box in a fast and effective way. UV-LED disinfection technology is an environmentally friendly and purely physical sterilization method, with no chemically corrosive components and it can quickly and effectively remove bacteria and other microorganisms from sterilized product boxes.

### 3.2.3. Induction Hand Disinfection Module

The disinfection product has an inductive hand disinfection module on the bottom of the product. When the hand enters the infrared sensing area, the quick disinfectant spray mode will be activated, disinfecting the hands without contact. The product is equipped with two nozzles, so there are two ways to dispense liquid, either as a spray or a lotion, suitable for 75% ethanol disinfection solution and no-rinse disinfection gel, to meet the different needs of liquid discharge.

### 3.2.4. Intelligent Sensor Technology Module

The product is equipped with an infrared sensor detection system with intelligent chip control. There are two infrared sensing areas. The first area is the front of the outer box of the disinfection product. When the hand brushes the side infrared sensor, the disinfection device will automatically turn on. The second area is the bottom of the disinfection product with another infrared sensor. When the hand brushes the bottom infrared sensor, it will open the quick disinfection liquid-spray mode which performs contactless disinfection of the hands and at the same time an infrared temperature measurement.

### 3.2.5. Intelligent Power-Off Technology Module

The product has the intelligent technology of self-power-off. When the children's UV disinfection product is not used for a long time, it will enter the automatic power-off function, saving energy and reducing consumption.

### 3.2.6. Intelligent Warning Technology Module

The product is loaded with infrared temperature measurement technology. Therefore, while reaching out for cleaning and disinfection, automatic temperature detection is performed, accompanied by a voice announcement.

When the human body temperature is higher than 37.3 degrees, the display light will change from green to red, enabling rapid screening of feverish people and improving the efficiency of epidemic detection and protection.

### 3.3. Three-Dimensional Visualization

After confirming the feasibility of the solution, a 3D printed model was created using MAYA 3D production software and a 3D rendering of the product was completed. Finally, some of the prototypes were sampled using light-curing 3D printing technology, as shown in Figures 2 and 3.

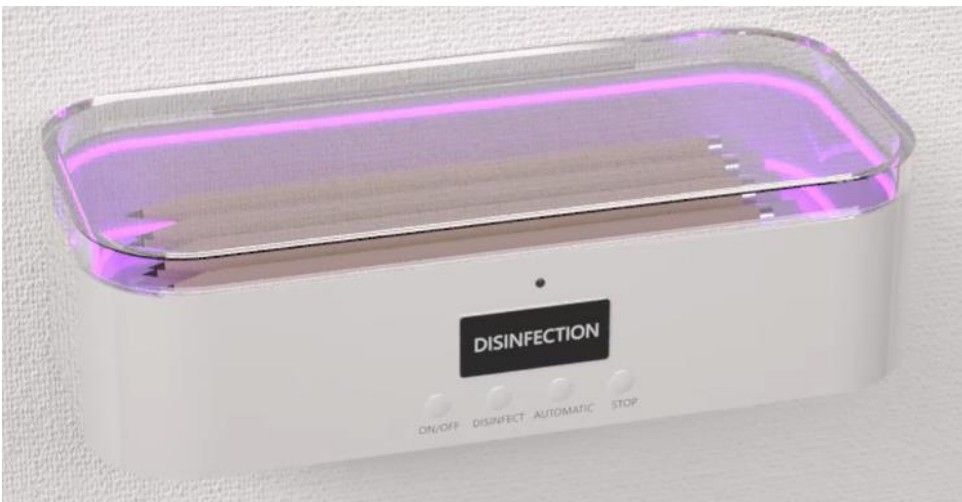

**Figure 2.** Prototype of a multifunctional induction UV disinfection product.

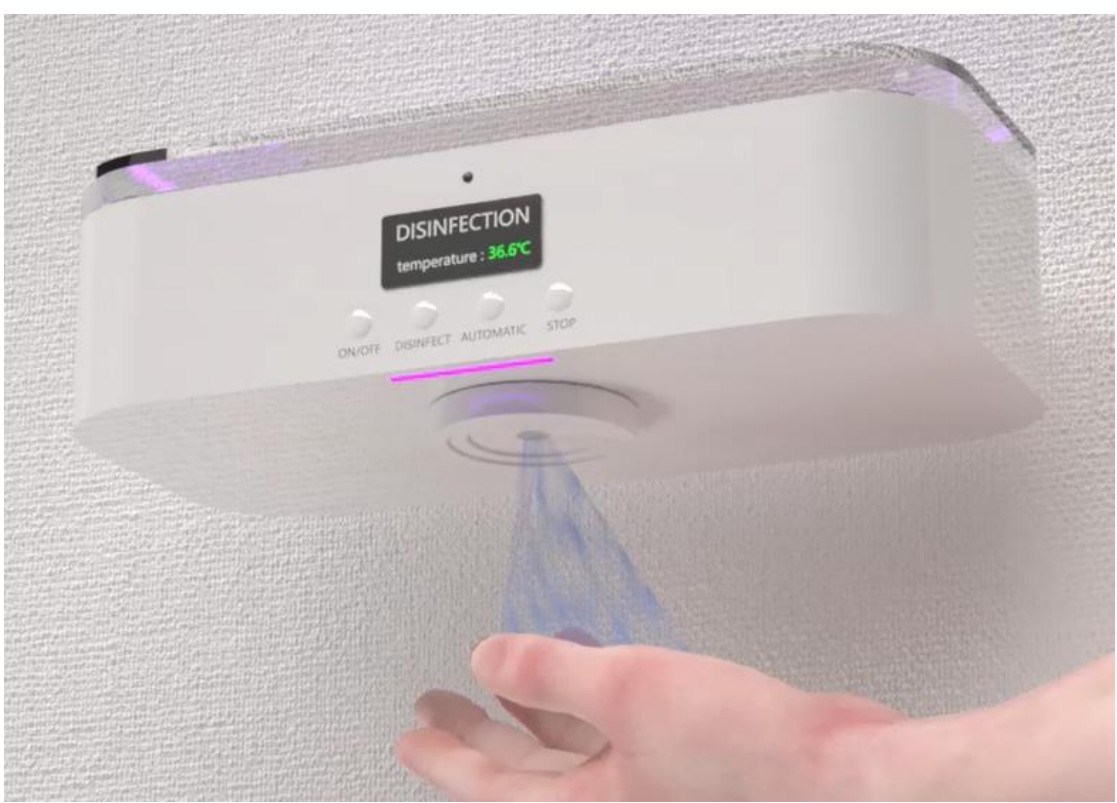

**Figure 3.** Multifunctional induction UV disinfection product hand disinfection use demonstration diagram.

## 4. Simulation Proof

### 4.1. Disinfection Effect Experiment

In this study, an ATP fluorescence detector was used to test the bactericidal effect of the product. McElroy found that firefly lanterns contained luciferin and enzyme luciferase which could be made to luminesce in the presence of ATP [17]. This finding could be utilized for the measurement of biomass or for the enumeration of microbial populations [18]. The yield information of ATP determination was similar to the measurement of traditional bacterial indicators of microbial contamination in terms of disinfection efficiency and control [19]. The ATP level detected on the surface of an object can therefore be used to infer the microbial level on the surface of the object and thus determine its hygiene status precisely.

The research team used an ATP fluorescence detector to detect bacterial content. The bacterial detection content before and after the use of the disinfection product were divided into the control group, and the bacterial situations of the control group were recorded and analyzed.

The experiments have shown that the bacteria levels after using this children's disinfectant product are much lower than before disinfection. This children's disinfection product has an extremely high kill rate against common germs and is fully capable of achieving what was previously envisaged.

### 4.2. Practical Assessment

In order to further ensure the practicability and effectiveness of the product, eight aspects were taken as the evaluation criteria based on the PUGH matrix judgment standard and judgment formula of the design scheme of the children's disinfection product—Reliability, Functionality, Operability, Efficiency, Friendliness, Trust, Environmental protection, and Aesthetics—and 10 primary and secondary school students were invited to use the product. The design scheme was compared with the existing disinfection scheme, and the PUGH matrix was obtained.

According to the comprehensive weight sequence E = {0.9298, 0.2411, 0.8822, 1.2716, 0.0476, 0.0888, 0.0663, 0.089} obtained before, the scores of each content were calculated,

and the comparison sequence was obtained, as shown in Table 9. It can be concluded from Table 9 that the score of this design scheme with eight aspects is much higher than that of the existing disinfection scheme, and the final comprehensive score is 5.3873, as shown in Table 9. The user's recognition is higher, indicating that the scheme can more effectively solve the current disinfection needs of school children and meet the design expectations.

**Table 9.** User-weighted post-matrix for a multifunctional inductive ultraviolet disinfection product for children.

| Evaluation Index (i) | Comprehensive Weight | User Evaluation for Scheme X3 with Weight | | | | | | | | | | Average Score |
|---|---|---|---|---|---|---|---|---|---|---|---|---|
| | | User1 | User2 | User3 | User4 | User5 | User6 | User7 | User8 | User9 | User10 | |
| reliability | 0.9298 | 0.9298 | 1.86 | 1.8596 | 0.9298 | 0.9298 | 1.8596 | 1.8596 | 1.8596 | 0.9298 | 1.8596 | 1.4370 |
| functionality | 0.2411 | 0.4823 | 0.4822 | 0.2411 | 0.4822 | 0.2411 | 0.4822 | 0.2411 | 0.4822 | 0.2411 | 0.2411 | 0.3507 |
| operability | 0.8822 | 1.7644 | 1.7644 | 0.8822 | 1.7644 | 0 | 1.7644 | 1.7644 | 1.7644 | 1.7644 | 1.7644 | 1.4436 |
| efficiency | 1.2716 | 2.5433 | 2.5433 | 1.2716 | 2.5433 | 1.2716 | 2.5433 | 1.2716 | 2.5433 | 1.2716 | 1.2716 | 1.8497 |
| friendliness | 0.0476 | 0.0476 | 0.0476 | 0 | 0.0476 | 0.0476 | 0.0476 | 0.0952 | 0.0476 | 0.0952 | 0.0476 | 0.0519 |
| trust | 0.0888 | 0.1777 | 0.0888 | 0 | 0.0888 | 0.0888 | 0.0888 | 0.0888 | 0.0888 | 0.1777 | 0.0888 | 0.0969 |
| environmental | 0.0663 | 0.0663 | 0 | 0.0663 | 0.0663 | 0 | 0.0663 | 0 | 0 | 0.0663 | 0 | 0.0362 |
| aesthetics | 0.089 | 0.089 | 0.089 | 0.178 | 0.178 | 0.089 | 0.178 | 0.089 | 0.089 | 0.178 | 0.089 | 0.1214 |
| total score | | | | | | | | | | | | 5.3873 |

## 5. Discussion and Conclusions

This paper investigates the practical teaching of a solar-energy-based UV disinfection product design for children, with the aim of providing additional services for school interventions to improve public health in primary and secondary schools. Three conclusions are drawn from the analysis of the solar-energy-based solution for the design of a multifunctional induction UV disinfection product for children:

(1) The creation and design of a new disinfection product fills a research gap in the disinfection of students' tools, strengthens graded disinfection strategies in schools, and reduces the potential risk of virus transmission in educational settings. Within the scope of this study, the disinfection effect of the product was fully tested and studied using an ATP fluorescence detector and sufficient proof of disinfection was obtained.

(2) Solar energy can be used to provide free green power for sterilization and disinfection devices, which is an infinitely renewable and zero-emission energy with no impact on the local environment.

(3) Throughout the research design process, AHP was used to determine the rationality of the program. The practicability and effectiveness of this multifunctional children's disinfection product have been verified by the experience of many experts and child users of the product. Advocacy and interactive research on how to increase the frequency of disinfection product use will be discussed in future work.

(4) This comprehensive practice model helps students to fully integrate technical means, artistic creation, and innovative practice, gives full play to the enthusiasm of students in the creative activities of social innovation practice, forms a good cycle, has positive teaching effects, and has practical significance.

**Author Contributions:** Writing–original draft, Q.X.; Supervision, X.W.; Project administration, Q.X.; Resources, B.Z.; Investigation, H.Z.; Methodology, J.C.; Validation, Z.Z. All authors have read and agreed to the published version of the manuscript.

**Funding:** The study was funded by the National Natural Science Foundation of China: 52075479, the National Social Science Foundation of Education Major Project: VFA220003, and the Department of Education of Zhejiang Province: 2022VPZGZ030.

**Data Availability Statement:** Data are contained within the article.

**Acknowledgments:** The authors would like to thank the associate editor and anonymous reviewers for their valuable comments and suggestions to improve this paper.

**Conflicts of Interest:** The authors declare no conflict of interest.

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
