# Peer review of "Digital Technology and Innovative Technology to Promote the Professional Development of Digital Media Based on Green Energy under COVID-19"

_processes, doi:10.3390/pr10101915_

Round 1

Reviewer 1 Report

The study presented in this paper is interesting and pertinent despite the decreasing problem of pandemic COVID-19.
The paper is well structured and written in a clear and objective language. The research objectives are clearly defined and, in view of the results presented, are effectively met.
The literature review is robust, with bibliographical references to relevant and up-to-date scientific publications.
Regarding future work, the authors are too vague. It is recommended to present in more detail the next phases of the study; as well as to clarify if the developed prototype will be introduced in schools or in the market. Will the solution designed in this study be limited to the academic research context or is its application and massification in a real context being studied?
In the title of the paper correct "Covid-9" to "Covid-19".

Author Response

Dear Professor, Thank you for your comments on our paper, we have learned a lot from it. We have examined the manuscript and revised it based on the comments after carefully studying your suggestion. All revised sections are shown in red in the manuscript. Attached is our response to the comment. If you need any additional information, please contact me via email immediately. My email is zjuxuqianqian@126.com. Thank you very much for your excellent and professional advice.
1. We adjusted the structure of the article and integrated the original introduction of the first part and the background of the second part to make it more concise and clear.
2. The Discussion and Conclusion sections have also been rewritten to make the results and conclusions more clearly presented.

Reviewer 2 Report

Citations should be checked. Likewise, it would be advisable to include current citations. The structure of the article should also be reviewed. Some sections are not clear. In this sense, some are too detailed and others, on the contrary, are too brief. Review the format of the tables to adjust them to the standards of the journal. Remove the 0 from before the . in decimal numbers. The discussion and conclusions section should be revised. In this case, the inclusion of citations that reinforce or refute the results obtained is missing. Check that the format of the figures complies with the instructions of the journal. Likewise, there is an excessive number of tables and figures that make reading difficult. For this reason, it would be advisable to assess which tables and figures should be left and which should be eliminated.

Author Response

Dear Professor, Thank you for your comments on our paper, we have learned a lot from it. We have examined the manuscript and revised it based on the comments after carefully studying your suggestion. All revised sections are shown in red in the manuscript. Attached is our response to the comment. If you need any additional information, please contact me via email immediately. My email is zjuxuqianqian@126.com. Thank you very much for your excellent and professional advice.

  1. We checked all citations, added citations, and deleted some unnecessary citations. At the same time, adjust the format of the citation to the standard of the journal.
  2. We adjusted the structure of the article and integrated the original introduction of the first part and the background of the second part to make it more concise and clear.
  3. We checked all the graphs and formatted them to conform to the journal's standards. At the same time, we have deleted some unnecessary tables and figures while ensuring the integrity of the data as much as possible.
  4. The Discussion and Conclusion sections have also been rewritten to make the results and conclusions more clearly presented.
